# Exploring variations in the implementation of a health system level policy intervention to improve maternal and child health outcomes in resource limited settings: A qualitative multiple case study from Uganda

**David Roger Walugembe**[1,2]*, **Katrina Plamondon**[2], **Frank Kaharuza**[3], **Peter Waiswa**[3], **Lloy Wylie**[4], **Nadine Wathen**[5], **Anita Kothari**[6]

1 Faculty of Medicine, Department of Anesthesiology, Pharmacology & Therapeutics, The University of British Columbia, Vancouver, British Columbia, Canada, 2 Faculty of Health and Social Development, School of Nursing. The University of British Columbia, Okanagan Campus, Kelowna, British Columbia, Canada, 3 Makerere University School of Public Health, Kampala, Uganda, 4 Schulich Interfaculty Program in Public Health, Department of Psychiatry, Pathology and Health Sciences, Western University, London, Ontario, Canada, 5 Arthur Labatt Family School of Nursing, FIMS & Nursing Building, Western University, London, Ontario, Canada, 6 School of Health Studies, Arthur and Sonia Labatt Health Sciences Building, Western University, London, Ontario, Canada

* dwalugembe@gmail.com, david.walugembe@ubc.ca

## Abstract

### Background

Despite growing literature, few studies have explored the implementation of policy interventions to reduce maternal and perinatal mortality in low- and middle-income countries (LMICs). Even fewer studies explicitly articulate the theoretical approaches used to understand contextual influences on policy implementation. This under-use of theory may account for the limited understanding of the variations in implementation processes and outcomes. We share findings from a study exploring how a health system-level policy intervention was implemented to improve maternal and child health outcomes in a resource limited LMIC.

### Methods

Our qualitative multiple case study was informed by the Normalization Process Theory (NPT). It was conducted across eight districts and among ten health facilities in Uganda, with 48 purposively selected participants. These included health care workers located at each of the cases, policy makers from the Ministry of Health, and from agencies and professional associations. Data were collected using semi-structured, in-depth interviews to understand uptake and use of Uganda's maternal and perinatal death surveillance and response (MPDSR) policy and were inductively and deductively analyzed using NPT constructs and subconstructs.

**Data Availability Statement:** The data supporting the findings of this study are associated with the doctoral thesis housed in the Electronic Thesis and Dissertation Repository at the University of Western Ontario. This repository provides a non-author point of contact for data access requests, as the full thesis from which this manuscript is derived is available there: https://ir.lib.uwo.ca/etd/8204/. Should further data or clarification is required, interested parties may contact either the University of Western Ontario's Electronic Thesis and Dissertation Repository or the authors directly, who will be pleased to assist with any additional requests.

**Funding:** The authors received no specific funding for this work.

**Competing interests:** The authors have declared that no competing interests exist.

## Results

We identified six broad themes that may explain the observed variations in the implementation of the MPDSR policy. These include: 1) perception of the implementation of the policy, 2) leadership of the implementation process, 3) structural arrangements and coordination, 4) extent of management support and adequacy of resources, 5) variations in appraisal and reconfiguration efforts and 6) variations in barriers to implementation of the policy.

## Conclusion and recommendations

The variations in sense making and relational efforts, especially perceptions of the implementation process and leadership capacity, had ripple effects across operational and appraisal efforts. Adopting theoretically informed approaches to assessing the implementation of policy interventions is crucial, especially within resource limited settings.

## Introduction

### Implementation of policy intervention to improve maternal and child health outcomes in resource limited settings

Ending preventable maternal and perinatal mortality is a global priority under the Sustainable Development Goals—2030 agenda [1]. The specific aim is to reduce the average global Maternal Mortality Ratio to less than 70 maternal deaths per 100,000 live births and neonatal mortality to at least 12 deaths per 1000 live births by 2030 [1–3]. Across low- and middle-income countries (LMICs), several interventions have been implemented to reduce maternal and perinatal mortality and improve health outcomes for mothers and babies. Despite the growing literature exploring implementation of these interventions and strategies, few studies have specifically focused on the implementation of policy interventions in these contexts [4–19]. Studies that have explored the implementation of non-policy interventions and strategies may be useful in understanding contextual influences on implementation of policy related interventions to improve maternal and child health [20–30].

Failure to achieve the Millennium Development Goals in many LMICs was attributed in large part to the inability to implement maternal and child health policies [12, 20, 31–33]. Previous studies have attempted to explore perceived challenges to implementing maternal and child health strategies, such as the World Health Organization maternal health guidelines to improve maternal and child care [7, 12, 14, 17, 19]. These guidelines, which address common challenges, include: augmentation of labour, induction of labor, prevention and treatment of postpartum hemorrhage and treatment of pre-eclampsia and eclampsia, breastfeeding, maternal and perinatal health guidelines, safe childhood checklist and task shifting in maternal and new born health [7, 14, 19, 34, 35].

### Use of theoretical approaches to study implementation of interventions to improve maternal and child health outcomes

While theory is an important contributor towards understanding the challenges to implementing maternal/child health guidelines and policies, few studies explicitly articulate their theoretical approaches [36, 37]. This may account for our limited understanding of the observed variations in implementation as well as their reported minimal impact in reducing maternal

and perinatal mortality and morbidity [10, 38–41]. Theoretical approaches provide a better understanding and explanation of how and why implementation succeeds or fails [10, 42–44]. Additionally, the use of theory to study the implementation of interventions offers generalizable frameworks that can apply across differing settings and individuals and the opportunity for incremental accumulation of knowledge as well as explicit frameworks for analysis [42, 45–48]. Helfrich et al. observe that using theory not only enhances understanding of barriers to implementation but may enhance the ability to design, and improve implementation processes [47].

Following a review of implementation science theories, models and frameworks, [42, 44], this study used the Normalization Process Theory (NPT) to explore what explains the variations in the implementation of MPDSR policy among selected health facilities in Uganda. As explained in detail elsewhere [49, 50], NPT combines the merits of multiple theoretical approaches with potential to offer a more complete understanding of certain aspects of implementation [51]. Its intent, level of abstraction, evidence of utilization in previous empirical studies, provision of how-to support tools and dual purpose as a theory and evaluation framework made it well-suited to the objectives of this study [43, 52–55].

## About Maternal Perinatal Death Surveillance and Response (MPDSR) policy

A maternal/perinatal death audit/review is an in-depth systematic review of maternal/perinatal deaths to delineate their underlying health, social and other contributory factors, and the lessons learned from such an audit are used in making recommendations to prevent similar future deaths [56, 57]. The WHO initiated the implementation of maternal and perinatal death reviews to go beyond the numbers captured by measures such as maternal mortality ratio (MMR) and infant mortality rate (IMMR) and facilitate understanding of the underlying reasons why women and their newborns die as well as devise contextually appropriate remedial actions [58, 59]. Kinney et al. noted that the implementation of these strategies has evolved from clinical obstetric to maternal death reviews (MDRs) and/or perinatal death reviews (PDRs), Maternal Death Surveillance and Response (MDSR) and currently Maternal Perinatal Death Surveillance and Response (MPDSR) [10]. Prior to 2012, much of the focus was on MDRs and/or PDRs. However, according to Smith et al. (2017), in 2012 the WHO and partners introduced the Maternal Death Surveillance and Response (MDSR) as a new approach aimed at collecting and using robust information for decision making [60]. Additionally, Kinney et al. observe that the integration of the perinatal death element into the approach was first reported in 2016 [10].

If implemented properly, maternal and perinatal death surveillance and response policy can reduce maternal mortality by up to 35%, and perinatal mortality by 30% [10, 36, 56, 57, 61–68]. However, despite recommendations by the WHO for its widespread implementation, there still exist variations in the implementation of MPDSR policy. For example, although 85% of LMICs have a national policy to review all maternal deaths, fewer than half are implementing MPDSR as per WHO guidelines [69]. In a recent survey of health facilities in four African countries, Kinney et al. (2020) indicated that fewer than half could provide evidence on any changes resulting from the implementation of MPDSR [70].

## MPDSR implementation in Uganda

As of 2020, Uganda was categorized among the 33 countries with high or moderate maternal mortality rate ranging between 100 to 499 deaths per 100,000 live births [1], but recent reports indicate that Uganda has made commendable progress in reducing maternal and neonatal

mortality rates. For example, according to the 2022 Uganda Demographic Health Survey (UDHS), Uganda's maternal mortality rate is reported to have reduced from 336/100,000 live births to 189/100,000 live births and the neonatal mortality rate has also reduced from 27/1,000 live births to 22/1,000 live births [71]. These gains have been partially attributed to the concerted efforts invested in the implementation of the MPDSR policy including collecting, reporting and analysing data on maternal and perinatal deaths. According to the MPDSR report for the year 2022/2023, strategies such as strengthening MPDSR coordination at the national and subnational levels, and the involvement of various stakeholders in responding to maternal and perinatal deaths were reported to have contributed to these successes [72].

Nonetheless, the implementation of the MPDSR policy still varies across settings within Uganda's health care system [72–74], and there is little research on Uganda's implementation of the MPDSR policy [50, 75–78], with very few of these studies explicitly articulating their theoretical approach [49, 50, 75]. Thorsen et al. (2014) observe that studies have focused on the entirety of the MPDSR process with heavy emphasis on establishing a committee and implementing the recommendations as a way to institutionalize them. However, understanding the implementation of policy interventions such as MPDSR requires examining both the processes involved and how the intervention becomes workable and integrated into everyday work [53]; there is a need to look at what people actually do and how they work. Embedding of a practice is dependent on organized and organizing agency [55] and requires continuous investment in sense-making, commitment, effort and appraisal of the routinization of a complex intervention [55]. It is against this background that we selected NPT for our study, whose primary research question was "what explains the variations in the implementation of MPDSR policy among selected health facilities in Uganda?" while the secondary research questions was "how do the actors involved in the implementation of MPDSR policy understand and make sense of the policy (coherence), engage, and participate in its implementation (cognitive participation), distribute work and resources among themselves to operationalize it (collective action) and reflect or appraise the effects of implementing the policy (reflexive monitoring)?"

## Materials and methods

### Study design and setting

This was a qualitative multiple case study informed by Stake (1995) and Merriam's (1998) approach to case study methodology [79, 80]. We adopted a constructivist paradigm [81, 82] to explore participant understanding of what explains the variations in the implementation of the MPDSR policy in eight districts in Uganda and among ten health facilities (cases) that were selected to represent four out of the seven levels of the Uganda health care system (health center III, health center IV, general hospitals and regional referral hospitals) (Fig 1- Structure /Arrangement of the Uganda Healthcare System). These four levels of care were mandated by the Ministry of Health (MoH) to implement the MPDSR policy effective 2008 [74]. Data were collected from 25 May 2018 and 30 March 2019. Sampling occurred at two levels: districts and cases within districts.

### Selection of districts and cases

The eight districts (refer to Table 1), were purposively selected for maximum variation to facilitate learning about a range of experiences MPDSR implementation [80]. The selection of the districts was informed by reviewing performance trends in district league tables [83] published in the annual health sector performance reports [84]. The reports also provided information on several health-related indicators, including the number of maternal and perinatal deaths per district local government from 2003/2004 [83].

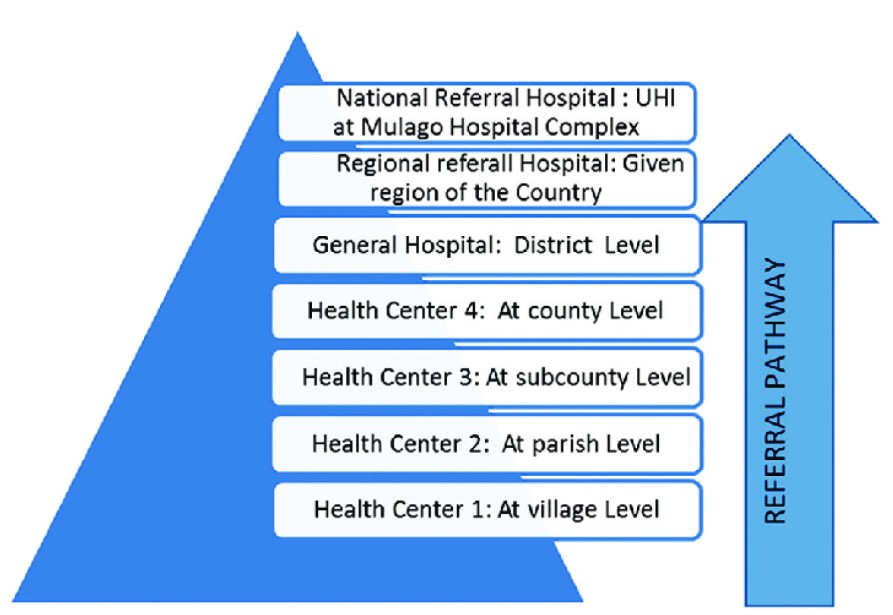

 Pediatric cardiovascular care in Uganda: Current status, challenges, and opportunities for the future - Scientific Figure on ResearchGate. Available from: https://www.researchgate.net/figure/Organization-of-Ugandas-Health-Care-System_fig1_312014998 [accessed 11 Jan, 2024]

**Fig 1. Structure /arrangement of the Uganda healthcare system.**

An initial six cases (D002, D003, D004, D005, D006, D008) were purposively selected from Jinja, Butambala, Mpigi, Mubende, Kibaale and Kabarole districts. At the recommendation of the MoH technical experts in charge of implementing the MPDSR policy, two cases (coded as D001 and D007) were added. These were selected from Kampala and Kasese district and represented the private-not-for profit sector. According to the technical experts, the inclusion of cases from this sector would enrich this study and would facilitate comparison between the initially selected cases, which were all government- funded health facilities. Since the two added cases were based in comparatively varying settings, that is urban (capital city) and rural, the decision to include both of them in the study was also aimed at understanding whether social contextual factors such as location contributed to implementation variations. Also, important to note is cases D052 and D062 (health center IIIs) were selected from within two districts (Mityana and Butambala) where two other cases representing a different level of care had previously been selected.

**Table 1. List of selected districts and number of cases per district.**

| District | Number of cases | Category |
|---|---|---|
| Butambala | 2 | Bottom Performing |
| Jinja | 1 | Top Preforming |
| Kabarole | 1 | Top Performing |
| Kampala | 1 | Top Performing |
| Kasese | 1 | Top Performing |
| Kibaale | 1 | Bottom Performing |
| Mityana | 2 | Top Performing |
| Mubende | 1 | Bottom Performing |

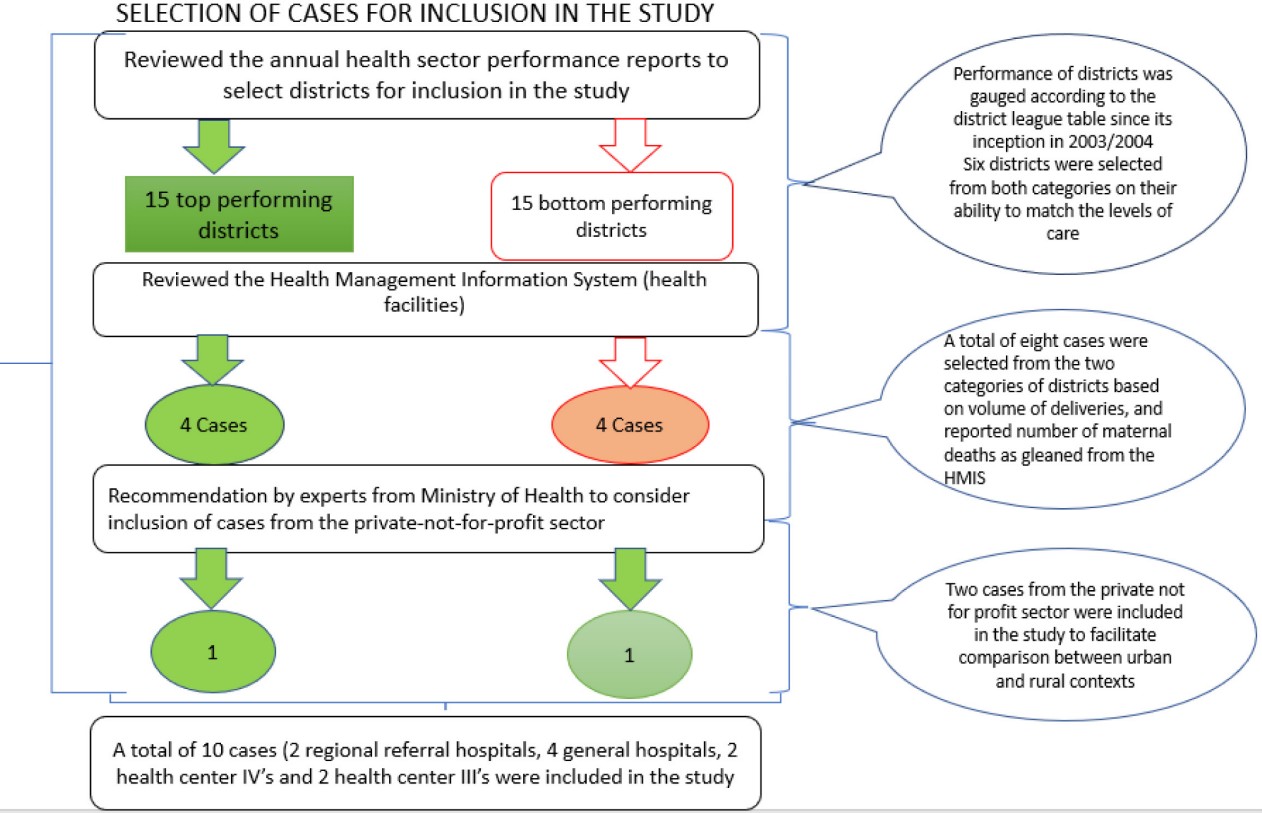

**Fig 2. Selection of cases from top and bottom performing districts.**

The selected districts consistently appeared among the 15 top (D001, D002, D005, D0052, D007, D008) and 15 bottom (D003, D004, D006, D0062) ranked local governments (performers) on the district league table. Matching a top with lower performing district was based on the level of care, volume of maternal deliveries conducted at each health facility as well as ownership of the health facilities. This same criterion was followed while selecting districts and cases representing other levels of care including general hospitals, health center IVs and IIIs (refer to Fig 2). Overall, 10 cases were selected based on their representativeness of the levels of care mandated by the MoH to implement the MPDSR [74]. Health facilities with the highest volume of maternal deliveries per district were included in the study. This was informed by the assumption that those with the highest volumes were more likely to have higher maternal and or perinatal deaths thus making them suitable case(s) for the exploration of the study questions. The volume of maternal deliveries was ascertained from a review of the district health information system (DHIS2) maintained at the Ministry of Health resource center.

## Selection of study participants

Study participants were purposively [85] selected from across the cases, the MoH, and from agencies and professional associations including the World Health Organization (WHO), United Nations Population Fund (UNFPA), Association of Obstetricians and Gynecologists of Uganda (AOGU), United States Agency for International Development (USAID) and the Uganda Health Service Commission (UHSC). The sampling was also informed by a review of

existing documentation and input from technical experts. All study participants were involved or should have been involved in the implementation of the MPDSR policy [58]. They were associated with a maternal and child health unit, department or national MPDSR committee and had worked in their current position for a period of not less than six months. This enabled them to suggest recommendations on strategies for addressing the causes of variations in implementation within their respective health facilities. Potential study respondents who were not in a position to discuss MPDSR policy or its implementation because of conflict of interest, or due to confidentiality agreements (n = 2), were excluded.

## Data collection procedures

In total, 48 people were interviewed, most at the respondent's place of work (n = 45). The semi-structured interviews lasted between 30 minutes -120 minutes. Interview guides (available on request) informed by the NPT constructs were used to guide the conduct of interviews with study participants (frontline health workers, administrative staff, representatives of agencies and professional associations and MoH staff) [52, 86, 87]. After the first two interviews, the guide was adjusted to increase clarity and respond to emerging findings (e.g., reflect language norms in context). With permission, interviews with study participants were digitally recorded.

## Data analysis

Interviews were transcribed, de-identified and archived in NVivo data management software. The first author (DW) read and re-read each transcript multiple times, to generate the initial codes, search for themes, and review, define and name them [88]. As a means for organizing text for subsequent interpretation, DW developed a codebook (Table 2) *a priori* informed by the NPT constructs and subconstructs [52, 86]. DW established how the emerging themes fit within the four major NPT constructs and their respective sub-constructs which were then respectively entered into NVivo as parent and child nodes. This process enabled the use of the theory to shape the potential interpretations of the emerging findings. DW also compiled analytical notes on the observed similarities and differences in the determinants of implementation of the MPDSR policy between the cases selected at each level of care and later between top and bottom performing districts (refer to Table 3). This analysis resulted into the identification of the broader themes that are presented below.

## Ethics approval and consent to participate

Ethics approval for this study was sought from the Health Sciences Research Ethics Board (HSREB, IRB 00000940) Delegated Review of the University of Western Ontario. Additional ethical approval was sought from the School of Medicine Research and Ethics Committee, Makerere University College of Health Sciences (REC REF No. 2018–018), the Uganda National Council for Science and Technology (HS 2393) and the Ugandan Ministry of Health (ADM 130/313/05). Participation in the study was completely voluntary and written informed consent was sought at all times. Study participants were assured of privacy and confidentiality and approved the use of information for improving public health, clinical practices and policy implementation. The manuscript does not include details, images, or videos relating to individual participants.

## Maintaining rigour

Method and data source triangulation were utilized to develop a comprehensive understanding of the variations in the implementation of the MPDSR policy [85]. Method triangulation

**Table 2. Codebook informed by NPT constructs and subconstructs.**

| Name of code (Parent node/construct) | Name of the subcode (child node/ subconstruct) | Explanation of the codes |
|---|---|---|
| Coherence | | The sense-making work that people do individually and collectively to operationalize the implementation of MPDSR (sense-making efforts). |
| | Differentiation | How implementing the MPDSR policy is different from other interventions aimed at improving maternal and child health. |
| | Communal specification | How people working together build a shared understanding of the aims, objectives, and benefits of implementing MPDSR. |
| | Individual specification | How participants collaboratively need to do things that will help them understand their specific tasks and responsibilities around the implementation of MPDSR policy. |
| | Internalization | How participants in sense making undertake efforts to understand the value, benefits, and importance of implementing the MPDSR policy. |
| Cognitive participation | | How people engage and participate in the implementation of MPDSR policy (relational efforts) |
| | Initiation | Whether or not the key participants are driving forward the implementation of MPDSR. |
| | Enrolment | How participants organize and reorganize themselves in order to collectively contribute to the work involved in implementing the MPDSR policy. |
| | Legitimation | Ensuring that other participants believe it is right for them to be involved, and that they can make a valid contribution. |
| | Activation | Actions and procedures needed to sustain implementation of MPDSR policy as collectively defined by the participants. |
| Collective Action | | Operational work that people do to enact MPDSR policy (operational efforts). |
| | Interactional workability | Interactional work that people do with each other and with elements of implementation of MPDSR when they seek to operationalize it in everyday settings. |
| | Relational Integration | Knowledge work that people do to build accountability and maintain confidence in the implementation of MPDSR and in each other as they implement the policy. |
| | Skillset Workability | Allocation of work that underpins the division of labour that is built up around implementation of MPDSR as it is operationalized in the real world. |
| | Contextual Integration | Resource work- managing implementation of MPDSR through the allocation of different kinds of resources and execution of protocols, policies, and procedures. |
| Reflexive Monitoring | | Appraisal work that people do to assess and understand the ways that MPDSR policy implementation affects them and those around them (appraisal efforts). |
| | Systematization | Determining the usefulness of implementing MPDSR for the participants and for the others and involves the work of collecting information in a variety of ways. |
| | Communal Appraisal | Working together-sometimes in informal collaboratives, sometimes in informal groups to evaluate the worth of a set of practices. |
| | Individual Appraisal | How individuals involved in the implementation of MPDSR work experientially as individuals to appraise its effects on them and the contexts in which they are set. |
| | Reconfiguration | Whether appraisal work by individuals or groups leads to attempts to redefine procedures or modify practices and to event to change/update the MPDSR guidelines and policy. |

[89] included the use semi-structured in-depth interviews and crosschecking accessible institutional records such as minutes from MPDSR review meetings, quality improvement reports, internal correspondence, DHIS2 databases and files of unprocessed/unreviewed maternal and perinatal death records. For data source triangulation, diverse categories of study participants were purposively selected and interviewed. Additionally, DW engaged in member checking after transcribing the audio-recorded interviews [79]. Furthermore, AK, NW, LW reviewed and commented on case findings as they emerged [79].

During data collection, DW maintained a database containing additional information collected from each case, including field notes manually taken during and at the end of each of interview. During data analysis, DW maintained a reflexive journal in which he noted reflections and ideas emerging from the interviews with the study participants. All these contributed

**Table 3. Summary of observed variations in the determinants of implementation across cases from the different levels of care.**

| Determinants of implementation | Level of care | | | | |
|---|---|---|---|---|---|
| **Sense-making Efforts** | Regional (Cases D004 & D008) | District- Public (Cases D005 &D006) | District- PNFP (Cases D001 &D007) | County (Cases D002 & D003) | Sub-county (CasesD0052 &D0062) |
| Perception | Mixed | Similar | Similar | Similar | Similar |
| Reasons | Mixed | Similar | Similar | Similar | Similar |
| Efforts | Mixed | Mixed | Mixed | Similar | Similar |
| Barriers | Mixed | Different | Similar | Different | Similar |
| **Relational Efforts** | | | | | |
| Leadership | Different | Similar | Mixed | Mixed | Similar |
| Organizational/structural arrangements | Mixed | Different | Mixed | Mixed | Similar |
| Legitimation strategies | Different | Mixed | Mixed | Mixed | Similar |
| Sustainability strategies | Different | Mixed | Mixed | Similar | Similar |
| Barriers | Mixed | Mixed | Mixed | Mixed | Different |
| **Operational Efforts** | | | | | |
| Teamwork and collaboration | Different | Different | Mixed | Similar | Similar |
| External networks | Different | Different | Different | Similar | Similar |
| Management support | Different | Similar | Similar | Different | Similar |
| Adequacy of resources | Different | Similar | Mixed | Similar | Similar |
| Barriers | Mixed | Mixed | Different | Mixed | Different |
| **Appraisal Efforts** | | | | | |
| Usefulness of intervention | Similar | Similar | Similar | Similar | Similar |
| Evaluation strategies | Similar | Mixed | Similar | Mixed | Similar |
| Effects on individuals | Different | Similar | Similar | Mixed | Similar |
| Adjustments/ reconfigurations | Different | Different | Different | Mixed | Similar |
| Barriers | Mixed | Mixed | Mixed | Different | Different |

to the development of the audit trail and enhanced the dependability of the study findings. Using a multiple case study design further enhanced the transferability of the study findings as the selected cases maximized diversity in the implementation of MPDSR policy which in due course may facilitate application of the study findings to other situations. Coupled with this, giving a thick description of the implementation of MPDR policy across various cases further facilitated transferability [79, 90].

## Results, discussion, and conclusions

This section provides thematic analysis of the variations in the determinants of implementation of the MPDSR policy across the top and bottom performing districts included in the case study. Descriptive characteristics of the selected cases and 48 interview participants are reported elsewhere [50], as in an account of how NPT was used to explore the variations in the implementation of the MPDSR policy [49].

Six themes regarding the observed variations across cases selected from both top and bottom performing districts were identified, including: 1) perception of the implementation of the MPDSR policy, 2) leadership of the implementation process, 3) structural arrangements and coordination, 4) extent of management support and adequacy of resources, 5) variations in appraisal and reconfiguration efforts and 6) variations in barriers to implementation of the policy. These themes were triangulated with responses from study participants selected from the Ministry of Health and Agencies and Professional Associations.

The determinants of the implementation of the MPDR policy and reported barriers were either different, similar, or mixed (had both similarities and differences) between the selected cases at the various levels of care. Different refers to circumstances where the determinants of implementation completely varied between the cases while similar refers to instances where the determinants of implementation were observed to be similar between the cases. Mixed refers to situations where the determinants of implementation between the cases had both similarities and differences.

## Theme 1. Perception of the implementation of the MPDSR policy as different or complementary

Most participants from both top and bottom performing districts perceived the implementation of MPDSR policy as being different from other interventions aimed at improving maternal and child health. A few participants, mostly from cases selected from the bottom performing districts, perceived the implementation of the policy as complementary to other such interventions. There were also participants from cases across both categories of districts that, despite being mandated to implement the MPDSR policy, did not know whether it was different from other interventions aimed at improving maternal and child health. These different understandings about policy implementation were shared by participants from the Ministry of Health as well as Agencies and Professional Associations. Participants felt the policy differed from others in its characteristics, negative outcomes and challenges encountered in its implementation process. In terms of characteristics, participants observed that compared to other interventions, the implementation of the MPDSR policy was time-bound, required more technical inputs as well as stakeholders. Additionally, unlike other interventions, MPDSR implementation was associated with negative outcomes such as the death of a mother, a baby or both. Finally, participants observed that MPDSR policy implementation encountered unique challenges such as blaming health workers, political interference, absence of a legal framework and indiscriminate access to medical records by the police, among others. These challenges are illustrated in the following quote:

> *We are still grappling with the difficulties of implementing the MPDSR. One, because it has become political, it has become legal, it is stressful for the health worker because it ends in death. . . . We have interpreted it to mean that the health worker who was looking after this mother or this neonate has committed a crime. So that already is difficult in itself, it brings a lot of problems. And you can see the health workers how they perceive it already being blamed and therefore its implementation cannot be completely smooth. While they would have perceived it as a quality-of-care improvement tool which it is, the rest of the people do not regard it like so. There is a lot of political interference and leadership interference when a death occurs. The sad thing is that they interfere at the time of death not at the time of ensuring quality of care. . . So really, I must say this has been one of the hardest policies to implement. Police coming and accessing information which it is not supposed to take away from the health facility unless under very strict conditions. . .But when a death occurs, the health worker becomes the victim. So, protection, we are grappling with protecting the health worker who is working within a broader weak system, health system.*

> *(APA 002F)*

However, those who perceived MPDSR implementation as complementary to similar interventions observed that it was part of a continuum that shared similar aims, objectives, context, approaches, and strategies to improve maternal health outcomes:

*Maternal and perinatal death reviews is one of the few packages in the whole complex of policies or approaches. . . I may not have a very good taste of how the others have played in the whole pool, but they contribute to one another, they are supposed to operate in a synergistic way. So, MPDSR is just an approach of making sure no mother dies.*

*(MoH R003M)*

The observed variations in perception of implementing the policy may further be explained by the lack of exposure to the policy among participants from cases that were meant to be implementing the policy but were not yet doing so. This lack of exposure in part was attributed to the limited ability of the Ministry of Health to distribute and disseminate the policy guidelines across the country, as noted by one study participants below:

*So, as we have the guidelines in place now, we have the challenge of rolling the guidelines across the country. We have over 120 districts and we have to institutionalize; we'll have to institutionalize these guidelines in all the districts. It needs a lot of capacity at least to bring on board the health workers, to have the committees in place, and also, we need some logistics to print the materials, distribute the funds to be able to effectively respond to the gaps that have been identified.*

*(MoH R001M)*

As a result of the above differences in the perception of implementation, there were variations in the levels of implementation effort invested across cases. For example, compared to the cases that perceived the implementation of the policy as being complementary and those that were not yet implementing, the cases that perceived the implementation as being different were observed to invest more effort via leadership, building and sustaining a community of practice, operationalizing, and appraising the effects of implementing the policy as discussed further below.

## Theme 2. Leadership of the implementation of the MPDSR policy

The difference in leadership efforts at the various levels of care was the second broad theme identified. For example, the leadership of the implementation efforts across all cases selected from the top performing districts were reportedly driven forward by multidisciplinary teams of frontline health workers and were supported by senior managers. Additionally, the majority of these teams reported having designated MPDSR focal persons, departmental or unit heads who ensured that the policy and related activities were implemented and that action points were followed through. On the other hand, the leadership efforts among cases selected from the bottom performing districts differed, with some reportedly driven forward by multidisciplinary teams of frontline health workers and senior managers while others were either led by only frontline health workers or were nonexistent. The observed differences in the investment of leadership efforts across all cases positively and negatively affected other efforts such as those aimed at legitimizing participation, sustainability, teamwork and collaboration, management support and allocation of resources for implementation. For example, cases with engaged and supportive leaders reported having more cohesive teams and invested more efforts in enacting feedback from the MPDSR meetings. Furthermore, those with active leaders reported encountering limited barriers to their efforts to legitimize participation of team members. As a result of these legitimation efforts, MPDSR implementation was more sustained at cases with active leadership compared to those with challenged leadership. This point was further

articulated by participants from both Agencies and Professional Associations as well as Ministry of Health, as highlighted below:

> . . . *Be it in terms of notification, in terms of making meetings come on, in terms of making sure that these meetings gradually take place, in terms of working on the recommendations, it still comes back to leadership. At the facility level, are the managers willing, so, are those units willing? Still, it comes back to us. You can invest, and train, and do what but people will not do it. So, it still comes back to the issues of leadership and also interest and also that passion at the individual and also at the facility level.*

*(APA 003M)*

> *You need strong leadership at the Ministry of Health headquarters who would one, ensure that these guidelines are universally adopted by all health institutions, private not-for-profit, private health facilities, public [health] facilities, all central institutions, all local governments. So, you need strong leadership there which encourages that. Then at the level of facility, they need to develop champions, that means you identify people, so it's not people just being recommended and say now we have an audit committee or review committee, you need to identify leaders who emerge, who are passionate to become champions and like to see this move forward. . . .*

*(APA 005M)*

### Theme 3. Structural arrangements and coordination of MPDSR policy implementation efforts

Our third theme pertains to the difference in structural arrangements and coordination of implementation efforts as a source of variation in implementation. Most of our cases reported having structures such as MPDSR Committees and District Health Teams (DHTs) that supported coordination and implementation. However, the duration of their existence, composition, functionality, availability, and level of support received from implementing partners, Ministry of Health and the community/local leaders greatly accounted for the differences in how these structures affected the coordination and implementation efforts. For example, while most of the cases from top performing districts reported having multidisciplinary and relatively functional MPDSR Committees and DHTs that were also supported by implementing partners, the composition, functionality, and support for these structures among cases from the bottom performing districts varied. While at some cases, the MPDSR Committees and DHTs had recently been established (D006), elsewhere they were reportedly not functioning adequately, lacked internal cohesion and teamwork (D004, D003) or were nonexistent (D0062). Compared to cases from the top performing districts, all structures across bottom performing district cases were not supported by implementing partners. In addition, due to the staffing challenges reported among these cases, the composition of the MPDSR committees did not follow MPDSR guidelines. Who was engaged also varied: the private-not-for profit case selected from the rural setting reportedly engaged stakeholders including the district health team, community leaders and politicians in its MPDSR Committee coordination and implementation efforts, but the case from the urban setting relied more on the health facility frontline health workers, senior managers, and hospital administration team. Indeed, participants from Agencies and Professional Associations and MoH further illuminated the importance of structural arrangements and coordination efforts on implementation success. For

example, despite acknowledging the extensive support (technical, financial, and material) that agencies and professional associations extended towards the coordination and implementation of the policy, the diverse number of implementing partners, their interests and priorities were reported to hinder the ability of the MoH to coordinate and equitably distribute them across the country.

A representative of one of the Agencies highlights the role of implementing partners at the national, district and health facility levels:

*So, we strengthen the government both at national and facility [level] but also districts to make sure that MPDSR is up and running. At national level we support coordination by Ministry of Health, there is a National MPDSR Committee to make sure it functions well, to compile, collect and analyze MPDSR data at national level which is an aggregate of information which comes from the districts. Then we also support the country or Ministry of Health to make sure that MPDSR committees at district but also facility, are in place. . ..*

*(APA 003M)*

However, coordinating these partners and ensuring equitable resource distribution across the entire health system were also observed as key challenges:

*. . .Then, because the players are so many, the coordination becomes a challenge for all of them to be brought on the same table to discuss the same subject at the national level but also at the district level, even at the health facility level because you may find that these different partners who are supporting particular intervention sometimes each of them brings in special needs because they want to capture different types of data. Although government definitely is very positive on this to say we have a national guidance on the kind of data, basic data we want to pick for our health information management system, first fulfill this and then the health worker can do other things.*

*(APA 002F)*

*No, we've been implementing but piecemeal. Different implementing partners and even the government have been implementing without a coordinated and harmonized approach but now with the guidelines we find it has been followed up because it centered on the establishment of a maternal [and] perinatal deaths surveillance committee at the facility and at district level such that we could be able to coordinate the notification and reviews. We could also mobilize resources in a coordinated manner, we could bring on board all the different stakeholders that is the policy makers, the implementers, and any other stakeholders in the community.*

*(MoH R001M)*

Additionally, the generally inadequate resources from the Government of Uganda for the health sector generally and MPDSR implementation specifically were also reported to subject the MoH to donor/development partner dependency (conditional funding), subsequently affecting structural arrangements and implementation coordination efforts:

*Ideally government is supposed to determine but partners will come with multisectoral funding and the problem of multisectoral funding, while they bring money, they try to use the principle of "he who pays the piper", at some point, for me I have funds and I want my funds to be put in the same place. So, while its multi-partner funding, they will ask government to identify*

*where they need to put the money. So, government will have to develop an index, given their strategic objectives of interest. So, you might find still that while most of the projects are packaged in such a way that the funder wants to implement a package in the same place, the different sectors will not have a big opportunity to determine as sectors where they want this funding to go, . . .That limits on how implementation is actually done and distribution of the services. . . .*

*(MoH R003M)*

And as further noted, this leads to vertical programming, duplication of administrative processes and interventions covering certain communities but not the entire health system.

*. . .While partners have been supporting all these processes, we have had them confined themselves to certain communities of their interest. . .like marginalized communities and war-torn zones, humanitarian communities, and this limits the implementation of the whole set of interventions in the whole country. So, you might find some part of the country has a dose of intervention well-articulated and well implemented but the other parts of the country where the problem could be, is not because they use more of an index decision. . ..*

*(MoH R003M)*

## Theme 4. Extent of management support and adequacy of resources to implement the MPDSR policy

Related to the above, our fourth theme describes how the differences in management support and adequacy of resources may explain the differences in implementation. Whereas cases from the top performing districts (except D0052) reported having supportive and engaged management that enabled policy operationalization, the extent of management support among cases from the bottom performing districts varied. For example, while participants from case D006 reported having supportive and engaged management, those from D003 were divided about the extent of management support. Among the most frequently reported parameters for gauging management support were active participation of managers in the maternal and perinatal death reviews, enacting feedback, and recommendations, demanding accountability, creating an enabling environment and resource allocation:

*Management support, of course now they have regular meetings, and they are able to address anything. In fact, weekly meetings. Two, they make sure that the ambulance and the driver is facilitated every day. They give supper to the driver, and they give some allowance to the driver for every day he sleeps around and then fuel is always there. And there is a hospital ambulance line which is now known to the community and all health centers have it such that in case they call it, this person should be able to respond. If this person is not able to respond they call one of the administrator's line because there are some people who know the administrators. There are times when the ED drives the ambulance if the driver is stuck somewhere, the ED goes there to pick the patient. Therefore, I think the management is somehow committed to seeing work done.*

*(D001 R002M)*

However, despite acknowledging management support, the majority of the participants across cases were divided about the adequacy of resources to operationalize the policy. For

example, whereas participants from one of the private-not-for profit cases (D007) observed that implementation does not require many resources and therefore had adequate resources, their counterparts from the rural case (D001) were divided about the same issue. While some said the resources were adequate others noted that the resources were inadequate. A similar trend was observed among cases selected from the public sector from both top and bottom performing districts. The contrast between these cases and observations about adequacy of resources even among participants from the MoH, Agencies and Professional Associations warrant further exploration of what resources are required to implement the MPDSR policy and at what levels. As local case respondents from an urban (D007) and rural (D001) case noted:

*For the audits, our audits are relatively cheap, the space is readily available, and we do not have to rent the space. Two, the meetings are early in the morning, so you do not need to provide break or lunch. So, there is no extra snack required and the meeting venue is relatively small, so you don't need a public address [system]. Then the documentation is manual, so you don't need a lot of computers. You may need like once in a while a projector and a laptop to screen if it's like a CME or something. That is all that would go into, like something you would really need but it's under the department, to make sure every department has a computer and an LCD projector. So, in terms of really whether we need a lot of resources to conduct an audit, no we don't.*

(*D007 R001M*)

*If I am to talk about neonatal, we are lacking a lot of resources because there is a NICU, we are supposed to have equipment, so we find out that we are lacking equipment. . . We find out like neonates are supposed to be fed by syringe drivers, we don't have. When we receive prematures we get stuck where to put them, incubators, only two warmers, only two. So, if they are more than four, we just improvise, and fix and yet cross infection comes in. So, when it comes to feeding materials, like now we are improvising, they are trying to buy raw plates and improvise. . . . So that is also a challenge. When it comes to bed codes, we are still lacking because we can accommodate 15–20 but when we get more than twenty, it becomes a problem. And that child labor [ward] in a month, we can admit about 70 to 100.*

(*D001 R004M*)

## Theme 5. Variations in appraisal and reconfiguration of the MPDSR policy implementation efforts

The fifth potential explanation for the observed variations in the determinants of implementation was in regard to the level of efforts invested by cases in assessing and understanding the ways in which the implementation affects them and others around them. Regardless of whether they belonged to the top or bottom performing districts, all cases that were implementing the policy acknowledged the usefulness of its implementation and cited examples of this. Among these were reduced maternal and perinatal deaths, improved service delivery, enhanced vigilance, improved staff performance, reflexive learning, and enhanced teamwork. To evaluate the worth/usefulness of implementing the MPDSR policy, participants reported using various approaches and strategies. These included continuing medical education sessions (CMEs), conducting maternal and perinatal death reviews/audits, meetings, implementing quality improvement projects, documentation and reporting as well as stakeholder

engagement activities such as radio talk shows and community outreach. Additionally, Agencies, Professional Associations and MoH noted performance ranking, conducting pilots and trainings, annual joint reviews, piloting an event tracker, establishing a call center as well as compiling an annual MPDSR report as implementation appraisal mechanisms. At the individual level, participants from across the cases reported that the implementation had both positive and negative effects although others observed that it no longer affected them since it had been routinized in their settings. Among the positive individual effects were enhanced vigilance and confidence, reflexive learning, commitment, motivation, mentorship, and quality improvement. Negatively, participants observed that the implementation increased their workload, led to exhaustion and delays in attending to patients. Some noted that implementation led to constrained interpersonal relationships where giving challenging feedback was misconstrued as blaming.

Across both categories of cases, participants reported undertaking infrastructural developments and readjustments as a result of implementing the MPDSR policy. However, with the exception of a nursery that was established at one of the cases from the bottom performing districts and the neonatal intensive care units (NICUs) that were established across both categories of cases, the majority of infrastructure changes were reported among cases selected from top performing districts. These included establishment of a maternity theater, high dependency unit (HDU), oxygen plant, a blood bank as well as a private wing at one of the cases. Whereas cases from the top performing districts cited more examples of service delivery related adjustments, the majority of the cases from the bottom performing districts were optimistic that the implementation of the policy would lead to more adjustments in the future. For example, while cases from the top performing districts reported hiring more staff, appointing focal persons, re-allocating staff, engaging volunteer nurses, adjusting MPDR forms, changing meeting times, providing accommodation to staff, acquiring equipment, and strengthening ambulatory services, only one case from among the bottom performing districts reported that policy implementation had resulted in the establishment of an MPDR committee and designing admission forms to capture vital observations.

Although at the health facility level participants were able to appraise and articulate the contribution of implementing the MPDSR policy as described above, most MoH, Agency and Professional Association participants noted that assessing its contribution on health outcomes such as mortality reduction was complex. They noted that the system-wide nature of the policy exposes it to several confounding factors which in due course limit attribution of outcomes to the policy itself. Despite the observed attribution challenges, participants noted that if well-implemented and institutionalized, benefits a could be measured and teased out:

> But overall, when it becomes part and parcel of a routine quality improvement, the facility will when you go to it, you'll find that there are many other good practices they have adapted. The health workers for example will develop performance plans as individuals, as part of good human resource management practices. You will find that they bond more as a team... You can't have the maternity side alone improving, the obstetric side alone improving, it's impossible, you cannot sustain it. So, that is why it's got to be the health systems improvement in general.
>
> (APA005M)

## Theme 6. Variations in the barriers to the implementation of the MPDSR policy

Overall, the existence of various barriers—the last theme—across different levels of care was an important driver of the differences in the implementation of the MPDSR policy. Although these barriers differed across cases, levels of care, and categories of study participants, there were some common ones worth noting. For example, across cases from the top and bottom performing districts, the most common reported barrier was limited or no feedback from the MoH regarding implementation of the maternal and perinatal death reviews. This was followed by inadequate knowledge and skills of the MPDSR processes, specifically the classification and reporting of deaths. The lack of systematic documentation approaches and inadequate materials were reported across cases although mostly among those from top performing districts. Additionally, inadequate staffing, funding, and supplies including blood shortages and drug stockouts, late referrals plus busy schedules or competing priorities were highlighted across both categories of cases. The structural set up of the health facilities was in some cases (from both categories) reported to constrain interdepartmental linkages and teamwork, along with a lack of a culture of data utilization, accommodation for frontline health workers, and inadequate equipment and ambulatory services. Although participants from top district cases highlighted inadequate performance incentives and the negative effects of the punitive approach that was adopted at the onset of implementing the policy as their unique barriers, participants from cases among the bottom performing districts cited several others. These included limited management support, negative staff attitudes, lack of teamwork, backlogs of unaudited maternal and perinatal deaths, infrastructural barriers such as inadequate space, lack of accessibility to and by some hard-to-reach communities, absence and or exit of implementing partners as well as poor information communication and technology infrastructure.

The existence and effects of these barriers on the implementation of the MPDSR policy was validated by accounts of participants from MoH, Agencies and Professional Associations, who also made recommendations on how to address them. For example, the limited or lack of feedback from the MoH as a common barrier affecting implementation of the MPDSR policy across cases was in part attributed to absence of a strong feedback and response mechanism to trigger immediate action at the national level:

*We don't have a mechanism for such a kind that tracks actions, . . . that responds adequately, maybe it's limited due to the inadequate resources that we have, that we may not have enough capacity to respond the issues as they come by. But if I am evaluating the system, that part is still missing. While they have tried to report and review, they may not have done the complete total reporting of all the deaths, but the numbers of maternal deaths that are reported are really scaring enough for us to have done something. I know it is because of limited resources because some work is being done definitely, there is a lot of work being done to respond to some of those issues, a lot of trainings have gone on to respond to the issues, but we do not have such a kind of impulsive or system to trigger an immediate response, the immediate response is still missing. The response might come but it might come later.*

*(MoH R003M)*

Additionally, the inadequate knowledge and skills of the MPDSR processes such as classifying and reporting maternal and perinatal deaths were also reported to hinder implementation across cases from all districts. As case-level providers reported:

*Now it happens that of course sometimes it becomes very tricky, where a mother comes in. This mother maybe has been mismanaged in a health center, if I can use that word, or maybe it is the patient or the attendant. There, the delays are not with us, then a mother comes in and within a short period of like maybe for you, you have managed for one hour and the mother dies. Now they count that maternal death that D005 has lost mother and the mother has to be counted on D005. That thing is still a challenge I must say, a very big challenge to us and of course you can't run away from it.*

(*D005 R004F*)

*But for the babies it is a bit tricky. So, for the perinatal [death], it is hard to know, I cannot judge but previously you would not even see them reporting. Like they have actually had either fresh still birth or a death before like between 28 to term, like still birth like they could not indicate more so the neonatal deaths, that death between 1 day to 7 days and that death that maybe the baby died in utero but in what, in that period of delivery. So, you would only get reports of macerated stillbirths, these deaths that happen sometime prior to what, to delivery. So, it is still a question, you would wonder actually why they were not getting it. But now they are a bit positive, when they have lost a baby, you would get a report.*

(*D006 R001F*)

Although the inadequacy of systematic documentation approaches, tools and materials was reportedly being addressed by implementing partners, participants also observed that it was affecting the quality of data as well as dissemination of success stories:

*But like we have discussed earlier, most of us in the hospital are in clinical service and you know we need to have someone step out of clinical service, and look and say, what are you guys doing? Are you doing it right? Are you doing it at the right time? Are you doing it the right way? And keep it up, you understand? We keep doing things and unfortunately, we lack that part in our [system]. . . so apart from a couple of people who come up with a publication and the like, but management is also tired of that. So, we are in a way trying to support getting someone to take a bird's eye view of the activities in the hospital. . . .. But I haven't seen a publication come out with how many positive pap smears have we seen? How many abnormal paps have we treated? When we treat these paps in D007, how many come back with after 5 years positive lesions? . . . So that is one of the challenges we have, that there is no one to take that look for us.*

(*D007 R002M*)

Furthermore, the negative staff attitudes reported mostly by participants from cases selected among the bottom performing districts were attributed to the negative experiences they had encountered while implementing the policy including experiencing blame for deaths, interrogation by police and at times interdictions from work:

*The attitude of health workers, implementing MPDSR is a big issue. It's not all health workers that have an attitude, but some health workers might have an attitude on MPDSR because of the previous experiences and maybe what they have seen others have gone through. For me that has really impacted so much on how people are doing MPDSR.*

(*MoH R003M*)

Beyond the common barriers that focused on health facilities, participants also highlighted limited access and utilization of the health facilities as major barriers affecting MPDSR implementation. The inability of mothers to access health facilities due to distance, high poverty levels coupled with inequitable distribution of health facilities was reported to hinder effective policy implementation to prevent deaths of mothers and babies from previously identified causes. However, as observed below, mothers and babies continue to die from avoidable causes, casting doubt on the usefulness of the MPDSR policy:

*The other challenge has been our communities being very rural and poor- poverty remains a very big issue. We have a limited number of[health] facilities that can provide both health and other social services, especially in geographical coverage. You might find that a number of [health] facilities could be there but most of them are confined in particular groups of density in terms of geographical distribution, they are more dense in some places and other places are not covered. That causes a bit of a challenge in having interventions distributed evenly in the country and covering the wider community of people who would benefit from them.*

*(MoH R003M)*

## Discussion

This study identified six themes that explain the observed variations in the determinants of implementation of the MPDSR policy across selected health facilities in Uganda: 1) perception of the implementation of the MPDSR policy, 2) leadership of the implementation process, 3) structural arrangements and coordination, 4) extent of management support and adequacy of resources, 5) variations in appraisal and reconfiguration efforts and 6) variations in barriers to implementation of the policy.

Considering that the variations in perception of the implementation of the policy affected the levels of relational, operational and appraisal efforts invested by the policy implementers across high and low performing health facilities, there is need to invest more sense making efforts especially differentiating how the implementation of the policy differs and or complements other interventions aimed at reducing maternal and perinatal mortality. Investing in sense making efforts and creating more awareness about the characteristics that distinguish the MPDSR policy from other interventions, its benefits, aims and objectives, processes involved, activities, inputs and outputs, as well as implementation barriers and facilitators, may contribute to success. While previous efforts have been geared towards understanding the barriers and facilitators to implementations [50, 75–78] further studies aimed at differentiating implementation of the MPDSR policy from other interventions could enable implementers to devise context-appropriate mechanisms and strategies to enhance integration and embedding within specific contexts (e.g., district vs. health facility).

Similarly, relational efforts, especially leadership, had ripple effects on other determinants of implementation. For example, it was observed that cases that invested adequate leadership efforts including establishing multidisciplinary teams, legitimizing participation of other members, designating MPDSR focal persons and ensuring accountability, performed better at implementation of the policy than those who did not. The absence of engaged leadership did not only negatively impact relational but also operational efforts including collaboration among team members, division of labour, allocation of resources as well as the extent of management support. Previous studies have highlighted leadership as a key facilitator of MPDSR policy implementation [10, 14, 17–19, 37, 61, 64, 78, 91]. Therefore, making sufficient

investments in developing leadership capacity for MPDSR policy implementation may help to address several identified barriers including those relating to the structural arrangements and coordination of implementation efforts.

While developing leadership capacity for MPDSR policy implementation may be critical to addressing the reported structural arrangement and coordination barriers, tasking the leadership with the responsibility to institutionalize regular appraisal (reflexive monitoring) of the policy implementation efforts across all health facilities could constitute a key priority. It could for example enable the leadership to review the composition, functionality and support required by district and health facility level MPDSR Committees. At the national level, engaging in regular reflexive monitoring efforts may facilitate adoption of strategies to reconfigure the approaches used to activate implementation of the policy across different levels of care. This would help address some of the observed discrepancies in the policy documents which reported that the Ministry of Health mandated various levels of care to implement the policy [74], yet there were health facilities that were yet to even learn about the policy. As such, findings from this study highlight the need to engage in reflexive monitoring efforts including undertaking a national assessment of all implementation efforts. However, undertaking such an assessment requires leadership and management support starting at the health facility level through to the Ministry of Health.

Relatedly, the reported absence of implementing partners across some of the health facilities mandated to implement the policy further points to the need for leadership to champion integration of equity considerations in the implementation of the policy [92]. Embedding equity considerations in the implementation of the MPDSR policy would ensure prioritizing equitable allocation of implementing partners, resources and supports across all health facilities mandated to implement the policy. It would also ensure paying attention to the observed contextual variations and associated negative consequences. For example, integrating equity considerations would help ensure that the plight of health facilities located in hard to reach parts of the country and their unique barriers that contribute to high maternal and perinatal mortalities in such contexts are considered and prioritized when allocating resources and support. It could also help to facilitate the engagement of the intended policy beneficiaries to contribute towards generating contextually appropriate strategies to reduce maternal and perinatal mortalities. Most importantly, embedding equity considerations could provide platforms for meaningful and equitable partnerships between the Ministry of Health and the implementing partners. Collectively, these actions could greatly contribute to enhancing the sustainability of the policy implementation efforts [93–95].

Whereas the challenge of inadequate resources remains endemic across most LMIC contexts and subsequently affects the capacity of health systems to engage implementing partners and strategically influence the agenda-setting processes [32, 33, 62, 63, 96, 97], there is need to explore the question of what resources are required to successfully implement the MPDSR policy across LMIC contexts, Uganda inclusive. This stems from the observed variations regarding the adequacy of resources between and among participants from both top and bottom performing categories. An enhanced understanding of the resources required to implement the MPDSR policy could help improve the capacity of national governments and institutions such as ministries of health and health facilities to plan and adequately manage the scarce resources. Additionally, it could influence adoption of strategies to sustain initiatives implemented courtesy of the support from implementing partners. Current evidence indicates limited continuity and sustainability of initiatives implemented courtesy of donor/implementing partner funding immediately after the initial funding runs out [93–95, 98].

Despite the reported ability to appraise the usefulness and benefits of its implementation, several challenges including absence of systematic approaches to measure and attribute the

observed reductions in maternal and perinatal mortality to the implementation of the MPDSR policy still abide [10, 61]. Coupled with this is the limited use of theories, models and frameworks as well methodologies to generate both empirical qualitative and quantitative data on the impact of MPDSR policy implementation on the reduction of maternal and perinatal mortalities [10, 49, 50]. Additionally, the inadequate systematic documentation approaches as well as the reported absence of a culture of data utilization hinder the appreciation of the intervention among key decision makers and the subsequent prioritization of decisions regarding resource allocation, staffing, accommodation for health workers as well as stocking supplies and consumables among other noted challenges [10, 61]. While the impacts of implementation can be tracked at health facility level, there is need to devise, and integrate standardized documentation approaches to systematically document and asses the usefulness and benefits of implementing the policy across the entire health system [10, 36, 37, 56, 57, 62–65, 67, 68]. Integrating theories, models and frameworks in the design, implementation and evaluation of MPDSR may as well enhance replication of efforts across similar settings and generalizability of findings [50, 61].

## Limitations

The lack of access to documented evidence to support claims by study participants who might have had an overrated perception of their implementation efforts could have constituted a limitation for the study. However, our qualitative data provide detailed descriptions of implementation processes beyond official documentation, with consistency in findings across the high and low performing districts. While the use of a theoretical approach helped to enhance the credibility and authenticity of the study findings, adjusting and customizing the language used to describe NPT constructs to effectively explore the study objectives was somewhat challenging. It resulted in several overlaps between participants' responses across various constructs and subconstructs complicating data coding and analysis. Nonetheless, an audit trail [99] of the study procedures followed from the stages of conceptualization, analysis to reporting is available [50]. Our data source triangulation [89], including field notes and member checks [79] further enhanced the credibility of the study findings.

Although private health providers were not covered in this study, findings from this study offer a more representative picture regarding the ongoing MPDSR implementation efforts and may inform larger scale/national assessments. The use of a multiple case study design further enhanced the transferability of the study findings as the selected cases maximized diversity in the implementation of MPDSR policy which may facilitate application of the study findings to other situations.

## Implications

First, this study contributes to substantive knowledge by providing much-needed evidence on what sense-making, relational, operational and appraisal efforts actors are currently investing in, and how these are affecting the implementation of the MPDSR policy across various settings. As noted by Kinney et al., although actors' subjective experiences, relationships, motivations, implementation climate and their ability to communicate greatly influence implementation processes, these have received little empirical attention [10]. As such, this study contributes contextually-relevant knowledge that may enhance understanding of the efforts invested by actors and subsequently help improve practice across the different levels of care. Additionally, uptake and utilization of these study findings by policy makers may enhance the capacity of the health system to address the high burden of maternal and perinatal mortality and contribute to strategies such as ending preventable maternal mortality to achieve sustainable development goals.

Second, findings from this study contribute to policy evaluation and organizational practice by providing a preliminary assessment of the implementation of the policy across various levels of care. Since the inception of the MPDSR policy in Uganda, there have not been empirical studies conducted across the health system to assess its implementation. Previous efforts have explored implementation of the policy in one facility or across health facilities funded by a particular implementing partner [75–78]. However, the design of this study has helped generate findings across various levels of care without being limited to a particular funding agency or implementing partner. Policy makers may thus utilize findings from this study to make evidence-informed decisions regarding current and future implementation efforts.

Third, findings from this study advance understanding of the utility of NPT in exploring implementation of health policy interventions. To the best of our knowledge, this is the first study to use an implementation science theoretical approach to explore implementation of the MPDSR policy across different levels of care of a health system. Previous studies have observed that the limited use of such approaches especially among LMICs may account for the limited understanding of the variations in the implementation of interventions including the MPDSR policy as well as their reported minimal impact in reducing maternal and perinatal mortality and morbidity [10, 38–42]. Furthermore, the use of NPT to study the implementation of the MPDSR policy may offer an explicit and generalizable framework for analysis of other implementation efforts across differing settings and individuals thereby enhancing applicability and utility of these study findings [42, 45–48].

## Conclusion

Across the top and bottom performing districts, the variations in MPDSR policy implementation were explained by differences in understanding of the implementation process, influence of cognitive participation efforts, the resource allocation and appraisal work around implementation of the policy as well as the differences in the barriers encountered in implementing the policy in different contexts. Broadly, findings from this study demonstrate that understanding implementation of complex policy interventions such as the MPDSR requires going beyond identifying the barriers and facilitators to exploring what the actors involved in the implementation processes actually do and how it affects implementation of the intervention. This can be accomplished through applying and utilizing theoretical approaches such as NPT. Exploring the sense-making, relational, operational and appraisal efforts that actors invest in implementation of complex interventions is critical to understanding why and how such interventions become embedded and sustained in some settings, yet struggle or completely fail in others. Insights generated from this study can be used to inform efforts to develop, design, modify and or scale up implementation of the MPDSR policy across the Ugandan health system as well as similar contexts.

## Supporting information

**S1 Checklist. Inclusivity in global research.**
(DOCX)

## Acknowledgments

The authors wish to thank the individuals who participated in the interviews and offered invaluable insights that facilitated making sense of the variations in the determinants of implementation of the MPDSR policy in Uganda.

## Author Contributions

**Conceptualization:** David Roger Walugembe, Frank Kaharuza, Peter Waiswa, Anita Kothari.

**Data curation:** David Roger Walugembe, Lloy Wylie, Nadine Wathen, Anita Kothari.

**Formal analysis:** David Roger Walugembe, Nadine Wathen, Anita Kothari.

**Investigation:** David Roger Walugembe.

**Methodology:** David Roger Walugembe, Frank Kaharuza, Peter Waiswa, Lloy Wylie, Nadine Wathen, Anita Kothari.

**Project administration:** David Roger Walugembe.

**Supervision:** David Roger Walugembe, Lloy Wylie, Nadine Wathen, Anita Kothari.

**Validation:** David Roger Walugembe, Lloy Wylie, Nadine Wathen, Anita Kothari.

**Writing – original draft:** David Roger Walugembe, Frank Kaharuza, Peter Waiswa, Lloy Wylie, Nadine Wathen, Anita Kothari.

**Writing – review & editing:** David Roger Walugembe, Katrina Plamondon, Frank Kaharuza, Peter Waiswa, Lloy Wylie, Nadine Wathen, Anita Kothari.

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
