## [Decision Letter · Decision Letter 0]

5 Aug 2024

PGPH-D-24-01046

Exploring variations in the implementation of a health system level policy intervention to improve maternal and child health outcomes in resource limited settings: A qualitative multiple case study from Uganda

Dear Dr. Walugembe,

Thank you for submitting your manuscript to PLOS Global Public Health. After careful consideration, we feel that it has merit but does not fully meet PLOS Global Public Health’s publication criteria as it currently stands. Therefore, we invite you to submit a revised version of the manuscript that addresses the points raised during the review process.

We look forward to receiving your revised manuscript.

Kind regards,

Laura Miniea Hoemeke, DrPH

Academic Editor

Journal Requirements:

Additional Editor Comments (if provided):

As the reviewers note, this is a timely manuscript and unique approach to policy analysis.

As Reviewer 2 notes, the different terms used is confusing. Health facilities are sometimes called "facilities," but also called "cases" or "sites." Systematically using one term--health facilities--would strengthen the manuscript. In addition, the term "study participants" is confusing, and I would suggest replacing it with "key informants."

Finally, a description of the policy would be helpful. It seems that the manuscript begins with a discussion of the MPDSR policy without a thorough definition and description of the policy. The section "About MPDSR" (line 185) provides a broad overview, and the section "MPDSR Implementation in Uganda" (line 209), but the authors do not describe the policy itself adopted in Uganda. Is there a measure of implementation, such as "all maternal deaths are systematically audited"? Clarification is needed, as different countries have different policies for the implementation of MPDSR.

Reviewers' comments:

Reviewer's Responses to Questions

**Comments to the Author**

1. Does this manuscript meet PLOS Global Public Health’s publication criteria? Is the manuscript technically sound, and do the data support the conclusions? The manuscript must describe methodologically and ethically rigorous research with conclusions that are appropriately drawn based on the data presented.

Reviewer #1: Yes

Reviewer #2: Yes

2. Has the statistical analysis been performed appropriately and rigorously?

Reviewer #1: Yes

Reviewer #2: N/A

3. Have the authors made all data underlying the findings in their manuscript fully available (please refer to the Data Availability Statement at the start of the manuscript PDF file)?

Reviewer #1: Yes

Reviewer #2: Yes

4. Is the manuscript presented in an intelligible fashion and written in standard English?

Reviewer #1: Yes

Reviewer #2: Yes

5. Review Comments to the Author

Reviewer #1: This study attempts to explain inconsistencies in rates of implementation of the MPDSR policy in various health systems in Uganda. The interview excerpts included in the manuscript highlight the underlying disparities between top-performing and bottom-performing facilities and lead to sensible suggestions to close the gap. Though the authors performed and analyzed a high number of interviews (n=48), the number of interviews per facility was relatively small considering 10 facilities were included. With such a small n value per facility or type of facility, each classification is highly susceptible to bias from the experience or viewpoint of individuals interviewed. It is a significant limitation of the study (and is acknowledged by the authors as such), but themes filtered from the data are important and uncover logical and long-standing disparities that demystify the policy's haphazard success rate across a single country and may provide insights on a larger scale.

Reviewer #2: This is an interesting and valuable study and NPT is an appropriate framework to use. However I feel at some points that the process of implementing the policy seems somewhat confused with implementing the recommendations of MPDSR itself. This is difficult as implementing the policy of conducting MPDSR is clearly really important and the first step in the process. However, each step is essential and the step most often missed and indeed the reason why MPDSR ulimately is at risk of failing to improve maternal and neonatal indicators is failure to deliver the "R" response. To clarify I feel it could be helpful to consider the implementation of the various components of MPDSR sequentially with a final focus on the implementation of the response component. Using this sequence to understand the points at which barriers arise would serve to clarify the issue.

Nevertheless, the theme that emerge are clear and of great significance in expanding our understanding of differentials in implementation.

I think it would improve the manuscript if you called "cases" "facilities" instead, as they clearly refer to health care facilities.

6. PLOS authors have the option to publish the peer review history of their article (what does this mean?). If published, this will include your full peer review and any attached files.

**Do you want your identity to be public for this peer review?** For information about this choice, including consent withdrawal, please see our Privacy Policy.

Reviewer #1: No

Reviewer #2: **Yes: **Helen Allott

---

## [Editor Report · Decision Letter 1]

29 Oct 2024

Exploring variations in the implementation of a health system level policy intervention to improve maternal and child health outcomes in resource limited settings: A qualitative multiple case study from Uganda

PGPH-D-24-01046R1

Dear DR Walugembe,

We are pleased to inform you that your manuscript 'Exploring variations in the implementation of a health system level policy intervention to improve maternal and child health outcomes in resource limited settings: A qualitative multiple case study from Uganda' has been provisionally accepted for publication in PLOS Global Public Health.

Best regards,

Laura Miniea Hoemeke, DrPH

Academic Editor